# Diagnostic Performance of PIVKA-II in Italian Patients with Hepatocellular Carcinoma

**DOI:** 10.3390/cancers17020167

**Published:** 2025-01-07

**Authors:** Valeria Guarneri, Elisabetta Loggi, Giuseppe Ramacieri, Carla Serra, Ranka Vukotic, Giovanni Vitale, Alessandra Scuteri, Carmela Cursaro, Marzia Margotti, Silvia Galli, Maria Caracausi, Lucia Brodosi, Filippo Gabrielli, Pietro Andreone

**Affiliations:** 1Department of Medical and Surgical Sciences, University of Bologna, 40138 Bologna, Italy; valeria.guarneri2@unibo.it (V.G.); giuseppe.ramacieri2@unibo.it (G.R.); alessandrascuteri@hotmail.com (A.S.); 2Operational Unit of Clinical Pathology, ASUR4, 63900 Fermo, Italy; bettaloggi@gmail.com; 3Interventional, Diagnostic and Therapeutic Ultrasound Unit, IRCCS AOUBO, 40138 Bologna, Italy; carla.serra@aosp.bo.it; 4Department of Emergency and Acceptance, General Medicine IV, University Hospital of Pisa, 56124 Pisa, Italy; ranka.vukotic@ao-pisa.toscana.it; 5Internal Medicine Unit for the Treatment of Severe Organ Failure, IRCCS AOUBO, 40138 Bologna, Italy; giovanni.vitale@aosp.bo.it; 6Department of Medical and Surgical, Maternal-Infantile and Adult Sciences, University of Modena and Reggio Emilia, 41126 Modena, Italy; carmela.cursaro2022@libero.it (C.C.); marzia.margotti2@unibo.it (M.M.); gabrielli.filippo@unimore.it (F.G.); pietro.andreone@unimore.it (P.A.); 7Microbiology Unit, IRCCS AOUBO, 40138 Bologna, Italy; silvia.galli@aosp.bo.it; 8Unit of Histology, Embriology and Applied Biology, Department of Biomedical and Neuromotor Sciences, University of Bologna, 40126 Bologna, Italy; maria.caracausi2@unibo.it; 9Clinical Nutrition and Metabolism Unit, IRCCS AOUBO, 40138 Bologna, Italy

**Keywords:** hepatocellular carcinoma, surveillance, diagnostic biomarkers, protein induced by vitamin K absence or antagonist II, alpha-fetoprotein

## Abstract

Hepatocellular carcinoma (HCC) is a leading cause of cancer deaths globally. Current gold-standard screening methods include imaging every six months and measuring alpha-fetoprotein (AFP) levels. This study evaluates Protein Induced by Vitamin K Absence-II (PIVKA-II) as a diagnostic biomarker for HCC and its correlation with disease stage. PIVKA-II levels progressively increase from chronic hepatitis C to liver cirrhosis and are highest in HCC patients, with levels correlating to advanced HCC stages based on the Barcelona Clinic Liver Cancer system. The optimal PIVKA-II threshold (>37 mAU/mL) achieved 80% sensitivity and 76% specificity, outperforming AFP, which had 53% specificity and 78% sensitivity at a cut-off of 5.2 ng/mL. Combining PIVKA-II and AFP improved diagnostic accuracy, achieving a positive predictive value of 73.9% and a negative predictive value of 94.2%. The findings suggest that PIVKA-II performs better than AFP alone and that their combined use enhances diagnostic accuracy, especially for advanced disease stages.

## 1. Introduction

Liver cancer is the sixth most common cancer worldwide, with an incidence of 865,000 new cases per year, and it ranks third in terms of mortality [1,2]. Among the different types of liver neoplasms, hepatocellular carcinoma is the most common primary liver cancer, accounting for 75–86% of all cases [3]. It is more frequently diagnosed in men than in women, with a male-to-female ratio of 2–2.5:1 [4]. The primary risk factor for HCC is cirrhosis, which can arise from various causes, including hepatitis B virus (HBV) and hepatitis C virus (HCV) infections, chronic alcohol abuse, and, more recently, metabolic-dysfunction-associated steatotic liver disease (MASLD, formerly known as non-alcoholic fatty liver disease or NAFLD), which is becoming increasingly prevalent in Western countries [5]. Furthermore, MASLD has become the dominant cause of HCC in the absence of cirrhosis [6].

The timing of diagnosis is crucial for therapeutic decision-making and significantly affects prognosis [7]. Patients diagnosed with late-stage HCC have an overall 5-year survival rate of less than 16%, whereas those diagnosed at an early stage have a 5-year survival rate exceeding 70% [8]. Early detection of HCC through surveillance methods in at-risk individuals has improved survival rates by enabling effective initial treatments, such as primary curative hepatectomy and locoregional ablative therapy [9].

Given the poor prognosis of HCC, optimizing the most suitable tools for achieving early diagnosis has become an ongoing challenge. Recent insights into the natural history of HCC have documented that it is primarily a slow-growing tumor, with its growth often paralleling that of the underlying cirrhosis. Cirrhotic patients constitute a high-risk group for the development of HCC and should participate in a regular surveillance program [10].

Based on these premises, surveillance is the most crucial step in diagnosing liver cancer and identifying additional tools for early diagnosis is essential for detecting tumors that are easier to treat.

Current guidelines recommend that abdominal ultrasound (US) should be performed every six months for the surveillance of cirrhotic patients. Additionally, the American Association for the Study of Liver Diseases (AASLD) guidelines suggest the optional use of a combination of US and of alpha-fetoprotein (AFP) [11,12].

Several biomarkers that can be easily measurable in serum or plasma samples have been proposed as appealing options due to their low risk, widespread availability, and ability to allow close monitoring without causing discomfort to patients. The best-known and most widely used surveillance marker in clinical practice is AFP, an oncofetal glycoprotein. Although AFP is not overexpressed in all patients with HCC, it is also present in other gastrointestinal tumors, pancreatic cancer, testicular cancer, embryonal carcinoma, and during pregnancy [13]. For this reason, there is an urgent need to identify biomarkers with enhanced specificity and sensitivity for early diagnosis.

Protein induced by vitamin K absence (PIVKA-II), also called des-gamma-carboxy prothrombin (DCP), is an abnormal form of prothrombin that differs from normal prothrombin due to the absence of 10 γ-carboxy glutamic acids. One of the crucial factors contributing to the increase in this marker is the enzyme γ-glutamylcarboxylase, which is dependent on vitamin K. In normal hepatocytes, this enzyme converts the prothrombin precursor into prothrombin, in particular by catalyzing the complete conversion of 10 glutamic acid (Glu) residues in the 10 γ-carboxyglutamic acid (Gla) domain of the prothrombin precursor in Gla residues, in the presence of vitamin K as a cofactor [14]. However, in HCC cells, γ-glutamyl carboxylase is defective [15,16]. The mechanisms underlying PIVKA-II production are not yet fully understood. Indeed, PIVKA-II is present not only in HCC tissues but also in non-cancerous tissues, such as in benign liver diseases like acute hepatitis and chronic hepatitis. Additionally, PIVKA-II levels can be elevated in various malignancies, including gastric cancer, pancreatic adenocarcinoma, and cholangiocarcinoma [17].

This marker is present in conditions of vitamin K deficiency or in patients treated with warfarin or phenprocoumon. Additionally, hypoxic conditions in HCC tissues contribute to the production of this marker [18]. These results suggest that PIVKA-II serves as a biomarker for the phenotypic status of HCC [19,20].

Evidence in the literature indicates that this marker is more sensitive and specific than AFP [17]. Notably, PIVKA-II levels increase in direct proportion to tumor size, demonstrating its ability to discriminate between early and advanced tumor stages. Furthermore, it is associated with extrahepatic metastases [21]. Recent data have also shown that elevated serum levels of PIVKA-II and its increased tissue expression are associated with microvascular invasion (MVI), a significant risk factor for recurrence and mortality in HCC [22]. In addition to these preliminary experiences, PIVKA-II has primarily been used in Asian countries. Despite its potential diagnostic value, the experience with this marker in Western countries, particularly in Europe, remains limited [23,24,25].

Based on this evidence, PIVKA-II could be evaluated as a diagnostic tool both for the surveillance of patients at risk and the diagnosis of HCC, including the early stages detection, as well as for monitoring patients undergoing treatment for liver cancer [26].

This study was designed to evaluate the role of PIVKA-II as a diagnostic marker of HCC by comparing serum levels of PIVKA-II and alpha-fetoprotein (AFP) in HCC patients and in two control groups, patients with chronic liver disease and cirrhosis without HCC. Furthermore, this study explored the possible correlation between the serum levels of PIVKA-II and the stage of HCC.

## 2. Materials and Methods

### 2.1. Study Design

This was a monocentric, cross-sectional study involving Italian patients with both primary and recurrent HCC in a real-life setting. Two control groups were analyzed: patients with liver cirrhosis (LC) and chronic hepatitis C (CHC). Participants were recruited from those attending the outpatient clinics at the Program of Therapeutic Innovation in Chronic Hepatitis at IRCCS AOU of Bologna, Italy, a clinical center highly specialized in the treatment of liver cirrhosis and HCC. The enrollment period began on 1 July 2017, with the inclusion of the first patient, and concluded on 16 July 2019, with the enrollment of the final patient. This study received approval from the local Ethics Committee (Comitato Etico Indipendente dell’Azienda Ospedaliero-Universitaria di Bologna, Policlinico S. Orsola-Malpighi, internal code 123/2017/O/Tess, approved on 16 May 2017) and all patients signed the informed consent forms for participating in this study and treating personal data.

### 2.2. Study Population

The study population consists of 80 consecutive patients with HCC (group A), 111 with LC (group B), and 111 with CHC who had never received antiviral treatment (group C). Patients were enrolled consecutively, and no predefined number of primary or recurrent HCC cases was established.

All patients met the following inclusion criteria: adult age (≥18 years); able to provide written informed consent; and only one of the following conditions: occurrence or recurrence of HCC diagnosed according to EASL guidelines (group A) or LC of multiple etiologies (in stage A or B of the Child–Turcotte–Pugh (CTP)) without HCC (group B), or untreated CHC (group C). Exclusion criteria included the following: pregnancy or nursing; HIV infection; any other neoplastic disease in addition to HCC; stages C and D of the Barcelona Clinic Liver Cancer (BLCL) staging system; prior or concurrent antiangiogenic treatment for HCC; use of coumarin anticoagulant drugs; and concomitant diseases that could interfere with laboratory analysis of the marker PIVKA-II, as determined by the Investigator.

### 2.3. Sample Collection and Marker Quantification

Group A (HCC): the blood sample was collected at the time of diagnosis, before the scheduled treatment.

Groups B and C (control groups): the blood sample was collected during an outpatient visit.

PIVKA-II levels were measured using serum immunological methods, specifically quantified by a microparticle chemiluminescence assay (CMIA) on an automated Architect platform ensuring a highly standardized, controlled, and reproducible system. The PIVKA-II measurement range is expressed in mAU/mL, with the reference range for the European population being 17.36 mAU/mL to 50.90 mAU/mL.

### 2.4. Statistical Analysis

The Kolmogorov–Smirnov test was used to assess the normality of data distribution. Differences between clinical groups were analyzed using the non-parametric Mann–Whitney U test or the Kruskal–Wallis test for multiple unmatched groups.

Correlations between variables were evaluated using the non-parametric Spearman rank correlation. Receiver operating characteristic (ROC) curve analysis was performed to assess the diagnostic accuracy of PIVKA-II and AFP as HCC markers, with the Youden Index used to determine the optimal cut-off.

A *p* value < 0.05 was considered statistically significant, with all tests being two-sided. Data handling and analysis were performed using SPSS software for Windows, version 17.0 (SPSS Inc., Chicago, IL, USA), and Prism Software, version 13 (GraphPad Software, Dotmatics, Boston, MA, USA).

## 3. Results

A total of 302 consecutive patients at different stages of liver disease, meeting the inclusion and exclusion criteria, were included in this study. The main demographic and clinical characteristics of the study population are summarized in Table 1 and Table 2.

The dataset is available only upon request to the first author.

The HCC and LC groups showed a similar distribution of liver disease risk factors with viral etiology accounting for the majority of patients in both groups, and approximately one-third presenting with multifactorial liver disease.

According to scoring systems for assessing disease stage and prognosis (CTP and Mayo End-Stage Liver Disease (MELD)), no patients in either group exhibited signs of advanced liver disease, and none had CTP C cirrhosis. Considering the 80 patients with HCC, 29 cases were newly diagnosed (de novo), while 51 were recurrent cases. In the HCC group, the BCLC staging system classified 25 patients (31.25%) as stage 0, 33 (41.25%) as stage A, and 22 (27.5%) as stage B. Moreover, in the analyzed cohort, the viremic state did not appear to significantly affect PIVKA-II levels (z score = 1.66557, *p* = 0.09492).

### 3.1. Distribution of PIVKA-II Serum Levels Across Different Comorbidities

In the overall study population, PIVKA-II levels ranged widely from 10.85 to over 2674.98 mAU/mL.

When stratified by study group, the distribution of PIVKA-II levels showed a progressive increase with the severity of liver disease, with significantly higher PIVKA-II levels observed in patients with HCC compared to those without HCC, including both LC and CHC patients (median [range min–max]: HCC = 67.19 [10.85–2674.98], LC = 30.95 [11.70–1250.54], and CHC = 24.89 [12.98–67.68]). Furthermore, patients with LC showed higher PIVKA-II levels than those with CHC (Table 3).

### 3.2. PIVKA-II Serum Levels in Relation to Nodules Number and Size

The mean PIVKA-II level was lower in patients with a single nodule compared to those with multiple nodules at the time of diagnosis (median mAU/mL [range min-max] multiple nodules 158.645 [10.85–2674.98], single nodule: 58.53 [19.27–989.02]; *p* = 0.719, z score −0.3607). Although a correlation exists between PIVKA-II levels and the number of nodules, it appears to be weak.

Regarding nodule size, PIVKA-II levels showed a significant correlation with the nodule dimension (*p* = 0.013, Person’s r = 0.235). However, when nodules were classified by size (less than 1 cm, between 1 and 3 cm, and above 3 cm), this significance was not observed (*p* = 0.194).

### 3.3. PIVKA-II Serum Levels According to HCC Type and Grading

The analysis of median PIVKA-II level showed no significant differences between de novo and recurrent HCC (*p* = 0.3843).

Due to the inability to collect HCC grading data for all patients, this analysis was conducted on 29 subjects, highlighting a linear correlation between PIVKA-II levels and HCC grading (*p* = 0.097).

### 3.4. PIVKA-II Levels in Relation to HCC Stage

To determine whether PIVKA-II levels are associated with HCC stage, patients were stratified according to BCLC stage, MELD score, and CTP.

PIVKA-II levels were significantly higher in BCLC-B than in BCLC-A (median mAU/mL [range min–max] BCLC-B: 234.935 [12.23–2674.98], BCLC-A 60.22 [10.85–1371.87]; *p* = 0.042), and also higher in BCLC-B compared to BCLC-0 (median mAU/mL [range min-max]: BCLC-B: 234.935 [12.23–2674.98], BCLC-0: 56.63 [25.92–299.6]; *p* = 0.041). No statistically significant difference was observed between PIVKA-II levels in BCLC-0 and BCLC-A (*p* = 0.352) (Table 3).

Regarding the MELD score, PIVKA-II levels appear to be moderately significantly correlated (Figure 1; Person’s R = 0.356, *p* = 0.001) and increase with higher MELD score.

On the other hand, PIVKA-II levels do not show a statistically significant difference among the various stages of the CP score (H score = 1.2558, *p* = 0.26244).

### 3.5. Analysis of Diagnostic Threshold for PIVKA-II

To assess and compare the diagnostic performance of PIVKA-II for the diagnosis of HCC, ROC curve analysis was performed. The area under the curve (AUC) was found to be 0.817 (95% CI: 0.755−0.887), with a cut-off value of 37 mAU/mL identified as optimal for maximize sensitivity and specificity as determined by the Youden Index (Figure 2).

This cut-off yielded sensitivity and specificity of 80% and 76%, respectively. Furthermore, when the established cut-off was applied to the study population, a moderate positive predictive value (PPV = 54.70%) and a strong negative predictive value (NPV = 91.35%) were observed (Figure 3). The diagnostic accuracy was found to be 77%.

### 3.6. Comparison of Diagnostic Thresholds for PIVKA-II and AFP

Further analysis was conducted to compare the diagnostic potential of PIVKA-II with that of AFP, the most widely used biomarker for HCC surveillance to date. Interestingly, the two biomarkers exhibited distinct behaviors across the entire study population, demonstrating a low degree of correlation (r = −0.007, *p* = 0.904). When examining only the HCC group, no correlation was observed between the two biomarkers (r = −0.049, *p* = 0.663).

The comparison of PIVKA-II and AFP as diagnostic tools for HCC revealed superior performance for PIVKA-II. Specifically, the AUC for AFP was 0.674 (95% CI 0.596–0.750, *p* < 0.001), which was significantly lower than the AUC of PIVKA-II (0.815; 95% CI 0.754–0.876 *p* < 0.001) (Figure 4).

The optimal cut-off for AFP was determined to be 5.2 ng/mL, yielding a specificity of 53% and a sensitivity of 78%.

The application of this cut-off resulted in a PPV value of 46.67% and NPV value of 81.82% (Figure 5).

In summary, elevated levels of PIVKA-II were observed in the majority of patients with HCC, but also in approximately one-quarter of patients without HCC, which diminishes its specificity.

Conversely, while AFP demonstrates robust specificity, exceeding the defined threshold exclusively in patients with HCC, it is abnormally expressed in only one-third of HCC patients.

Interestingly, 38 patients with HCC had AFP levels below the threshold. However, 30 of these patients (79%) exhibited PIVKA-II levels above the threshold.

### 3.7. Combination of PIVKA-II and AFP

Interestingly, the combination of both PIVKA-II and AFP yielded the highest diagnostic accuracy, achieving 73.9% and 94.2% for PPV and NPV, respectively. This combined approach enhances the diagnostic capability compared to the use of each marker in isolation (Figure 6).

### 3.8. Other Considerations

Other exploratory analyses were conducted in the subgroups, using an AFP threshold value of less than 20 ng/mL, as this is the most commonly used cut-off in clinical practice and guidelines.

The subjects with AFP <20 ng/mL were 54 out of 80 (67.5%); of these, 42 (77.8%) had a PIVKA-II >37.5 mAU/mL. Of the 22 subjects who had AFP >20 ng/mL, only 4 (18.2%) had a subthreshold PIVKA-II value. The subjects with the first nodule <3 cm were 63 out of 80 (75%); of these, 49 (77%) had a PIVKA-II value >37.5 mAU/mL. Of the 17 with volume >3 cm, 3 (17%) had a subthreshold PIVKA-II value.

Combining AFP and PIVKA-II, only 12 out of 80 HCC (15%) showed levels of AFP <20 ng/mL and PIVKA-II <37.5 mAU/mL, while 39 out of 80 (48.75%) showed both molecules over the threshold. Considering subjects with first nodule <3 cm, 10 subjects out of 80 (12.5%) showed both AFP and PIVKA-II levels below the threshold, while 12 subjects out of 80 (15%) showed both molecules above the threshold.

## 4. Discussion

This study aimed to elucidate the role of PIVKA-II as a diagnostic marker of HCC by comparing its levels in patients with HCC and in those with liver disease without HCC.

The dismal prognosis associated with HCC underscores the necessity of optimizing early diagnostic strategies to enhance the likelihood of achieving complete recovery.

The potential for utilizing multiple biomarkers to aid in the identification of HCC has gained a significant research interest in this field [2]. The advantages of incorporating biomarkers into routine clinical practice are clear: a serum-based biomarker, particularly when measured using automatic platforms, can facilitate the screening of at-risk populations in a manner that is easy to implement across various settings. This approach does not require sophisticated facilities or specialized staff expertise and it is both reproducible and relatively cost-effective. The rationale for this approach is not to replace the imaging techniques, particularly US, which are widely recognized as the most appropriate test for HCC surveillance according to the latest guidelines [2]. Instead, the aim is to support diagnosis and positively influence subsequent outcomes. Currently, serological tests under investigation for HCC include AFP, DCP (also known as PIVKA-II), and several others, or a combination of these biomarkers [21]. The use of combinations of serum biomarkers for predicting the development and early detection of HCC has shown promising results in several studies [27,28,29]. As recently reported by El-Serag et al., the integration of various biochemical markers reflecting different aspects of HCC pathogenesis has demonstrated good performance in predicting HCC development in patients with diverse liver diseases [27]. A recent proteomics-driven study by Xing et al. identified a panel of four biomarkers, including PIVKA-II, capable of detecting HCC and predicting which cirrhotic patients are at risk of developing HCC [29]. The AFP is the most widely utilized biomarker for HCC; however, its performance is suboptimal, given that not all HCCs express AFP. The detection rate of HCC that is not visualized by ultrasound (US) remains relatively low [2].

PIVKA-II has emerged as a promising biomarker for HCC, with accumulating evidence evaluating its diagnostic capabilities both independently and in combination with other biomarkers for HCC detection and post-treatment surveillance [21].

The results of the present study indicate that patients with HCC exhibit significantly higher levels of PIVKA-II compared to patients with LC or CHC, who do not have HCC. The elevation in the PIVKA-II levels correlate progressively with the disease stage, reflecting the natural history of hepatic disease. This finding corroborates previous data obtained for similar cohorts [26,30,31,32], reinforcing the robustness of this evidence. It is worth noting that our sample did not include patients with obstructive jaundice, a condition that has been reported to be associated with a potential increase in PIVKA-II levels [33].

In contrast to other studies, this research offers several advantages. Notably, it involves a real-life study population, as patients were recruited from a single clinical center and followed up regularly based on a limited set of eligibility criteria. This approach aims to avoid creating an “ideal” patient profile, thereby enhancing the generalizability of the findings. This prerequisite accounts for various multifactorial diseases and comorbidities within the study population. This study aimed to assess the diagnostic performance of PIVKA-II in a real-world setting, encompassing patients of all ages, with different histories and conditions. Additionally, the recruitment from a single clinical center ensured consistency in the study procedure and criteria for the definition of HCC. Another advantage of this study was the ethnicity of our population, which consisted of Caucasian subjects. To date, the vast majority of PIVKA-II data have been derived from Asian cohorts [19]. Given this more extensive experience, PIVKA-II is currently included in the evidence-based clinical guidelines of the Japanese and Asia Pacific Association for the Study of Liver (APASL) [34].

The cut-off identified in our study as the optimal threshold for detecting HCC, as determined by the Youden Index, is slightly higher than that proposed by Asian guidelines. However, it is important to note that the Youden Index, while widely used, has its limitations. It may not provide an optimal threshold for all clinical applications, as the choice of cut-off can depend on the relative costs of different types of clinical errors. Additionally, the identified threshold may not necessarily generalize to future studies or different patient populations. Other optimization methods or indexes could yield alternative thresholds that may be better suited for other specific contexts. However, it closely aligns with the cut-off found in a case-control study conducted on a French population [30] and in an Italian study [35]. This observation underscores the necessity for further investigation of European cohorts, as they may significantly differ from Asian patients.

Additionally, our study confirms that AFP and PIVKA-II are independent biomarkers, exhibiting no correlation [35]. Most importantly, these results demonstrate that PIVKA-II offers superior diagnostic performance.

Much evidence suggests that PIVKA-II and AFP should be utilized in combination to enhance sensitivity and specificity [35,36,37]. This study aligns with those findings. Consequently, while the combined use of these biomarkers improves both sensitivity and specificity, a significant number of patients remains undetected.

An important finding of this study is the association between PIVKA-II levels and the stage of HCC, as indicated by BCLC score, highlighting its potential implication for patient outcome and the prediction of survival and recurrence. These results are consistent with a recent study that focused on a cohort of patients with compensated or decompensated LC, both with and without concomitant HCC [32]. However, whether this aspect limits the use of biomarkers in the context of early-stage remains to be clarified. A larger cohort or a longitudinal study are needed to better define the predictive capabilities of PIVKA-II, either alone or in combination with other biomarkers.

When applying the cut-off of 20 ng/mL to our case series, as proposed in the literature [38], we observed lower specificity, as 70% of HCC patients reported a value below this threshold.

### 4.1. What Does the Study Add to Current Knowledge?

At present, PIVKA-II is not utilized in routine clinical practice, and its plasma measurement is not recommended in the current guidelines for the management of hepatocellular carcinoma. We believe that additional evidence, such as that provided by our study, may help experts justify its potential incorporation into clinical practice. Moreover, while studies on PIVKA-II are relatively abundant, there is a notable lack of research specifically focused on European populations, particularly in Italy, which adds further value and relevance to our findings.

### 4.2. Limitations of the Study

Our study had several limitations. First, the sample size was relatively small, and the study was single-center. Furthermore, the etiology of liver disease is heterogeneous, encompassing viral, alcoholic, dysmetabolic, autoimmune, and cryptogenic causes. This variability prevented us from determining whether the increase in plasma PIVKA-II levels may also be partially influenced by the underlying etiology of liver disease. Additionally, some patients presented with a first occurrence of hepatocellular carcinoma, while others had a recurrence of the tumor. Furthermore, the cross-sectional design of this study did not allow us to assess PIVKA-II’s performance to predict disease recurrence over time.

## 5. Conclusions

In conclusion, PIVKA-II levels are elevated in HCC patients compared to control groups without HCC. The levels of PIVKA-II correlate with the severity of HCC, indicating its potential as a more reliable biomarker. Additionally, PIVKA-II demonstrates superior diagnostic performance compared to AFP in screening for HCC within our patient cohort. Conversely, AFP alone proves to be inadequate as a screening test; however, its use in combination with PIVKA-II may enhance diagnostic accuracy, albeit still falling short of optimal performance.

## Figures and Tables

**Figure 1 cancers-17-00167-f001:**
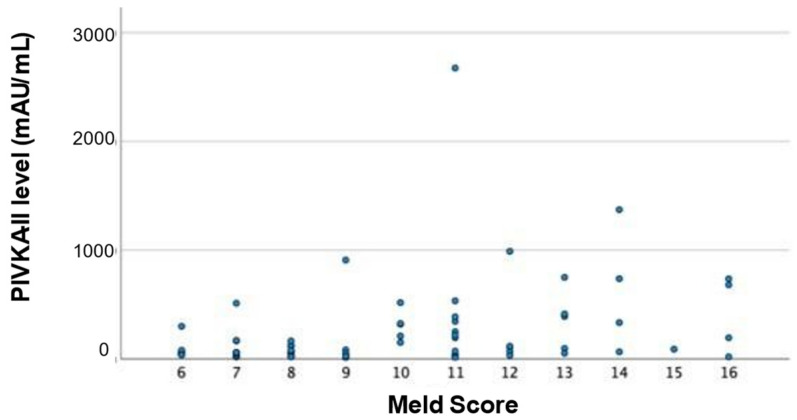
PIVKA-II levels mAU/mL in 80 patients with HCC, analyzed for MELD score each blue dot represents a PIVKA-II level.

**Figure 2 cancers-17-00167-f002:**
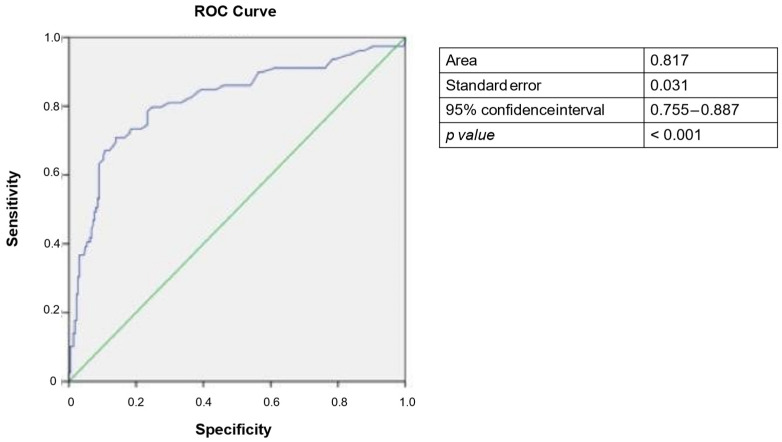
Receiver operating characteristic (ROC) curve for PIVKA-II as diagnostic biomarker for HCC. Blue line refers to PIVKA-II levels, green line is a reference.

**Figure 3 cancers-17-00167-f003:**
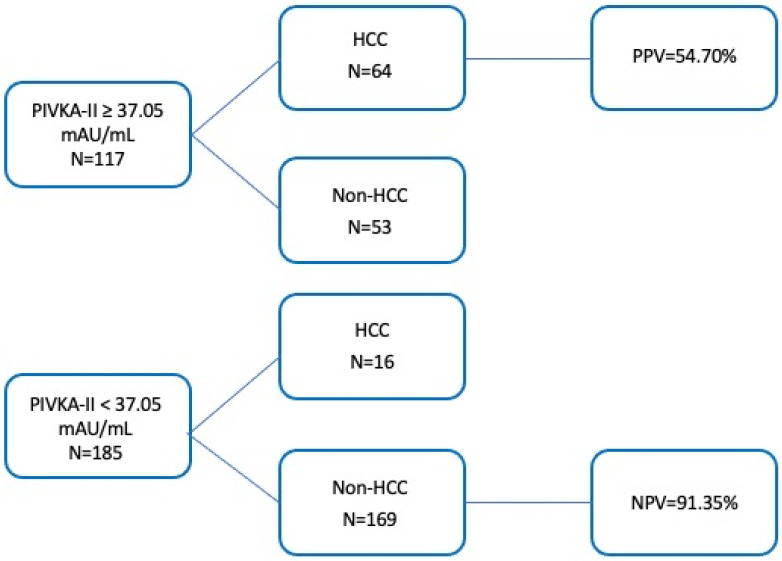
Predictive values of the PIVKA-II. Cut-off 37.05 mAU/mL on identification of HCC patients.

**Figure 4 cancers-17-00167-f004:**
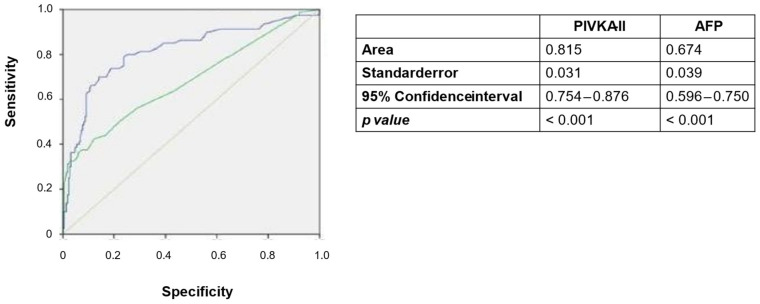
Receiver operating characteristic (ROC) curves for PIVKA-II and AFP as diagnostic biomarkers for HCC. Blue line = PIVKA-II; green line = AFP.

**Figure 5 cancers-17-00167-f005:**
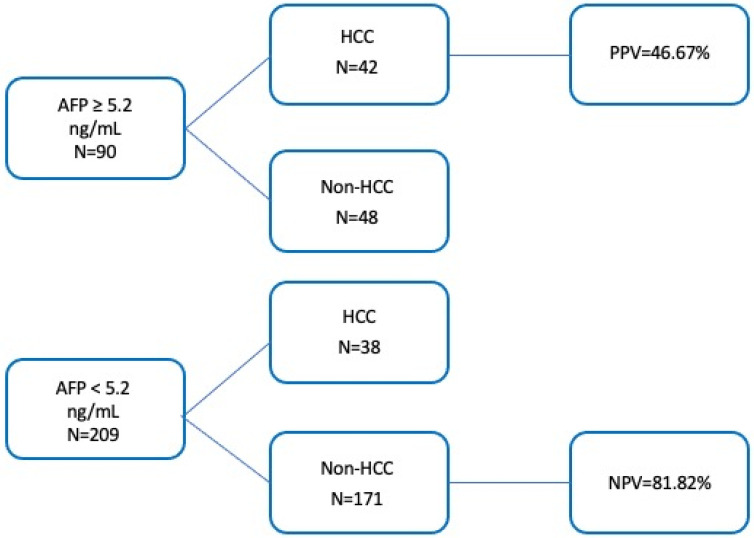
Predictive values of the AFP. Cut-off 5.2 ng/mL for identification of HCC.

**Figure 6 cancers-17-00167-f006:**
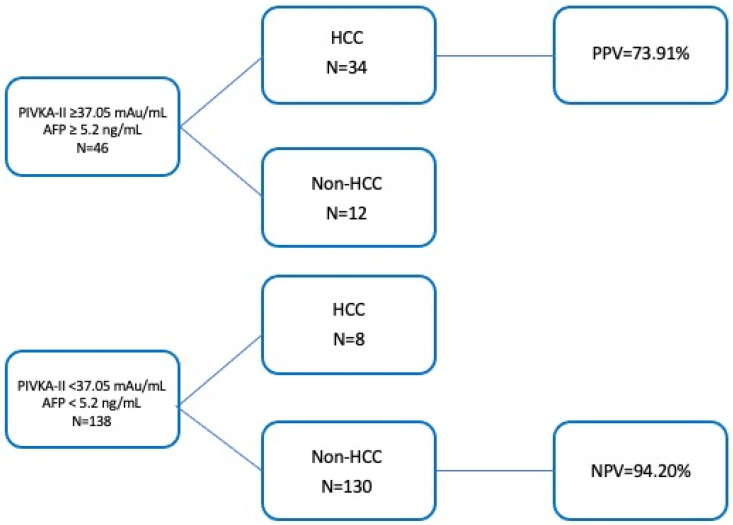
Predictive values of combination PIVKA-II/AFP at established cut-offs for identification of HCC patients.

**Table 1 cancers-17-00167-t001:** Baseline clinical and demography features of patients enrolled.

Variable	HCC *N* = 80	LC *N* = 111	CHC *N* = 111
Median Age (Years)	66 [48–85]	63 [20–83]	59 [28–84]
Male/female: N [%]	65/15 (81%/19%)	67/44 (60%/40%)	41/70 (37%/63%)
Liver Disease Etiology			
Hepatitis C infection	19 (23.75%)	46 (42%)	104 (94%)
Hepatitis B (±Delta Coinfection)	7 (8.75%)	11 (10%)	0 (0%)
Alcohol abuse	10 (12.5%)	5 (4%)	0 (0%)
NAFLD	4 (5%)	9 (8%)	0 (0%)
Cryptogenetic disease	2 (2.5%)	2 (2%)	0 (0%)
Multifactorial disease:			
Viral + Alcohol	2 (2.5%)	14 (13%)	1 (1%)
Metabolic + Alcohol	10 (12.5%)	5 (4%)	0 (0%)
Viral + Metabolic	3 (3.75%)	13 (12%)	6 (5%)
Metabolic + Autoimmune	1 (1.25%)	1 (1%)	0 (0%)
Mixed etiology *	23 (28.75%)	5 (4%)	0 (%)
CTP Score			
A	63 (78.75%)	102 (92%)	110 (99%)
B	17 (21.25%)	9 (8%)	1 (1%)
Meld Score	9 [3–16]	8 [1–22]	5 [0–9]
BCLC Score			
0	25 (31.25%)		
A	33 (41.25%)		
B	22 (27.5%)		

* More than three risk factors. HCC = hepatocellular carcinoma, LC = liver cirrhosis, CHC = chronic hepatitis C, CTP = Child–Turcotte–Pugh, N = number of subjects, NAFLD = non-alcoholic fatty liver disease, BCLC = Barcelona Clinic Liver Cancer. Categorical variables are reported as absolute number (% of total) while continuous variables are reported as median [min–max].

**Table 2 cancers-17-00167-t002:** Baseline clinical features of patients enrolled.

Variable	HCC *N* = 80	LC *N* = 111	CHC *N* = 111
Type 2 Diabetes Mellitus	30 (37.5%)	29 (26%)	8 (7%)
AFP (ng/mL)	5.5 [1.3–43,857]	3 [1–109]	3 [1–33]
ALT (U/L)	24 [9–173]	23 [4–225]	42 [10–277]
AST (U/L)	30 [11–175]	28 [12–174]	34 [13–232]
Total bilirubin (mg/dL)	1.03 [0.33–3.2]	0.96 [0.2–3.93]	0.7 [0.2–1.5]
Creatinine (mg/dL)	0.82 [0.44–2.57]	0.79 [0.47–2.6]	0.7 [0.4–1.2]
GRF (mL/min)	90.09 [26–117.24]	92.05 [22.45–134.29]	95.21 [51.07–126.41]
INR	1.135 [0.96–1.5]	1.17 [0.9–1.73]	1.01 [0.8–1.16]
Albumine (g/dL)	3.81 [2.7–4.7]	4 [2.74–4.8]	4.1 [3.4–4.8]
Neutrophils (n./mm^3^)	3.19 [1.01–8.58]	2.8 [0.96–7.99]	3.34 [1.3–10.6]
Lymphocytes (n./mm^3^)	1.345 [0.4–8.1]	1.46 [0.38–5.44]	1.92 [0.69–4.7]
Platelets (n.×10^9^/L)	113 [35–290]	108 [21–303]	229 [47–430]

HCC = hepatocellular carcinoma, LC = liver cirrhosis, CHC = chronic hepatitis C, *N* = number of subjects, AFP = alpha-fetoprotein, ALT = alanine aminotransferase, AST = aspartate aminotransferase, GFR = glomerular filtration rate, and INR = international normalized ratio. Absolute number (% of total) or median [min–max].

**Table 3 cancers-17-00167-t003:** Protein Induced by Vitamin K Absence-II (PIVKA-II) level distribution according to clinical condition, number of nodules and to hepatocellular carcinoma (HCC) Barcelona Clinic Liver Cancer scores. Mann-Whitney test was used to analyse two groups while Kruskal-Wallis test was used to compare three groups. LC: liver cirrhosis; CHC: chronic hepatitis C; 0 = HCC very early stage; A = HCC early stage; B = HCC intermediate stage.

Analysis	First Variable	Second Variable	z Score	*p* Value
Mann-Whitney	HCC	LC	−6.810	<0.001
	HCC	CHC	−40.235	<0.001
	CHC	LC	−40.235	<0.001
	Multiple nodules	Single nodule	−0.361	0.719
	0	A	−0.934	0.352
	0	B	−2.042	0.041
	A	B	−2.035	0.042
Kruskal–Wallis			H score	*p* value
	HCC-LC-CHC		474.269	<0.001
	0-A-B		5.738	0.057

## Data Availability

The dataset is available only upon request to the first author.

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
