# Peer review of "Diagnostic Performance of PIVKA-II in Italian Patients with Hepatocellular Carcinoma"

_cancers, 2025, doi:10.3390/cancers17020167_

Round 1
Reviewer 1 Report
Comments and Suggestions for Authors
In this review, the authors aimed to evaluate the diagnostic role of PIVKA-II in HCC. Their findings indicate that PIVKA-II levels correlate with HCC severity and demonstrate greater diagnostic accuracy compared to AFP. Furthermore, combining PIVKA-II with AFP enhances both sensitivity and specificity, achieving a positive predictive value of 73.9% and a negative predictive value of 94.2%. To make this manuscript suitable for publication, there are some comments pointed out for consideration.
1. The role of PIVKA-II in HCC diagnosis has already been investigated in a number of studies including western and eastern population. What are the advantages of authors’ findings?
2. The Introduction and Discussion contain too many paragraphs, making the key points unclear. It is recommended that the authors merge some of the paragraphs and reorganize the expression. In Introduction, certain background information that is not closely related to the study can be omitted to enhance focus.
3. The limitations of AFP are as follows: (1) Approximately 30% of HCC patients exhibit AFP negativity, defined as AFP <20 ng/mL (PMID: 35002508); (2) AFP has low diagnostic sensitivity in early-stage HCC (tumor diameter <2 or 3 cm). Therefore, it is recommended to perform a subgroup analysis in these two subgroups to evaluate the diagnostic performance of PIVKA-II and the combination of PIVKA-II and AFP.
4. In Discussion, it is recommended to add a paragraph analyzing the limitations of this study, such as its single-center design and relatively small sample size.
5. In Discussion, it is recommended to clarify whether there are patients with obstructive jaundice in this study. Studies indicates that serum PIVKA-II levels in such patients are above the normal upper limit (PMID: 1702578).
6. In Discussion, it is recommended to supplement several important papers, as follows: a longitudinal phase III study to validate a robust metabolic biomarker panel predictive of HCC (PMID: 38365278), the longitudinal assessment of serum biomarkers for HCC detection (PMID: 30153338), a mass spectrometry-based proteomics panel for early diagnosis of HCC (PMID: 38110372).
7. Previous studies have shown that the pathogenic mechanisms of HCC vary by etiology, leading to different levels of biomarker elevation (PMID: 39547439; PMID: 39547439). Please add an analysis in the study on how etiology affects the diagnostic performance of biomakers.
Author Response
Reviewer 1
In this review, the authors aimed to evaluate the diagnostic role of PIVKA-II in HCC. Their findings indicate that PIVKA-II levels correlate with HCC severity and demonstrate greater diagnostic accuracy compared to AFP. Furthermore, combining PIVKA-II with AFP enhances both sensitivity and specificity, achieving a positive predictive value of 73.9% and a negative predictive value of 94.2%. To make this manuscript suitable for publication, there are some comments pointed out for consideration.
- The role of PIVKA-II in HCC diagnosis has already been investigated in a number of studies including western and eastern population. What are the advantages of authors’ findings?
We sincerely thank the reviewer for requesting clarification on the innovative aspects of our study. At present, PIVKA-II is not utilized in routine clinical practice, and its plasma measurement is not recommended in the current Western guidelines for the management of hepatocellular carcinoma. We believe that additional evidence, such as that provided by our study, may help experts justify its potential incorporation into clinical practice. To further support this, we have included the dataset in the supplementary materials to encourage its use in future meta-analyses. Moreover, while studies on PIVKA-II are relatively abundant, there is a notable lack of research specifically focused on PIVKA-II in European populations, particularly in Italy, which adds further value and relevance to our findings. We inserted these considerations in the paper, at the end of the discussion.
- The Introduction and Discussion contain too many paragraphs, making the key points unclear. It is recommended that the authors merge some of the paragraphs and reorganize the expression. In Introduction, certain background information that is not closely related to the study can be omitted to enhance focus.
We thank you for the suggestion. We removed the paragraph on new diagnostic approaches involving PIVKA-II to make the introduction more concise.
- The limitations of AFP are as follows: (1) Approximately 30% of HCC patients exhibit AFP negativity, defined as AFP <20 ng/mL (PMID: 35002508); (2) AFP has low diagnostic sensitivity in early-stage HCC (tumor diameter <2 or 3 cm). Therefore, it is recommended to perform a subgroup analysis in these two subgroups to evaluate the diagnostic performance of PIVKA-II and the combination of PIVKA-II and AFP.
Thank you for your suggestion. We have conducted the requested analysis and incorporated the results into the text, in section 3.8: “Subjects with AFP < 20 ng/mL were 54 out of 80 (67.5%), of these 42 (77.8%) had a PIVKA-II > 37.5 mAU/mL. Of the 22 subjects who have AFP > 20 ng/mL, only 4 (18.2%) have a subthreshold PIVKA-II value. The subjects with the first nodule < 3 cm were 63 out of 80 (75%), of these 49 (77%) had a PIVKA-II value > 37.5 mAU/mL. Of the 17 with volume > 3 cm, 3 (17%) had a subthreshold PIVKA-II value. Combining AFP and PIVKA-II only 12 out of 80 HCC (15%) showed levels of AFP < 20 ng/mL and PIVKA-II < 37,5 mAU/mL while 39 out of 80 (48,75%) showed both molecules over the threshold. Considering subjects with first nodule < 3 cm, 10 subjects out of 80 (12,5%) showed both AFP and PIVKA-II levels below the threshold, while 12 subjects out of 80 (15%) showed both molecules above the threshold.”
A comment on the requested statistical analysis has been added to the discussion, along with the corresponding bibliographic citation: "Applying the cut-off of 20 ng/ml to our case series, as proposed in literature [PMID: 35002508], we observed lower specificity, as 70% of HCC patients reported a value below this threshold."
- In Discussion, it is recommended to add a paragraph analyzing the limitations of this study, such as its single-center design and relatively small sample size.
Thank you for the suggestion. We have added a brief paragraph at the end of the discussion: “Our study has several limitations. First, the sample size is relatively small, and the study is single-center. Furthermore, the etiology of liver disease is heterogeneous, encompassing viral, alcoholic, dysmetabolic, autoimmune, and cryptogenic causes. This variability prevents us from determining whether the increase in plasma PIVKA-II levels may also be partially influenced by the underlying etiology of liver disease. Additionally, some patients presented with a first occurrence of hepatocellular carcinoma, while others had a recurrence of the tumor. Furthermore, the cross-sectional design of the study does not allow us to assess PIVKA-II’s performance to predict disease recurrence over time.”
- In Discussion, it is recommended to clarify whether there are patients with obstructive jaundice in this study. Studies indicates that serum PIVKA-II levels in such patients are above the normal upper limit (PMID: 1702578).
Thank for suggesting this issue, we add in the discussion the reference with this comment: “It is worth noting that our sample did not include patients with obstructive jaundice, a condition that has been reported to be associated with a potential increase in PIVKA-II levels.”
- In Discussion, it is recommended to supplement several important papers, as follows: a longitudinal phase III study to validate a robust metabolic biomarker panel predictive of HCC (PMID: 38365278), the longitudinal assessment of serum biomarkers for HCC detection (PMID: 30153338), a mass spectrometry-based proteomics panel for early diagnosis of HCC (PMID: 38110372).
Thank you for the suggestion. We have added a paragraph on this topic in the discussion section.
- Previous studies have shown that the pathogenic mechanisms of HCC vary by etiology, leading to different levels of biomarker elevation (PMID: 39547439; PMID: 39547439). Please add an analysis in the study on how etiology affects the diagnostic performance of biomakers.
We have read your comment with interest, although the references you suggested do not align well with the content of our article. We chose not to include a statistical analysis investigating the variation of PIVKA-II across the different etiologies, as the sample of HCC patients is small (80 patients), and further dividing it by etiology would not yield statistically significant results.
Reviewer 2 Report
Comments and Suggestions for Authors
Dear Authors
The clinical study titled ‘Diagnostic Performance of PIVKA-II in Italian Patients with Hepatocellular Carcinoma’ was evaluated. The study was carried out with scientifically and multidisciplinary. I congratulate all the people who participated in the study. ​My opinions and suggestions regarding the study are given below. The study comply with scientific criteria and is suitable for publication.
1) The title of the study is appropriate
2) The people participating in the study are suitable for team ethics. The study was conducted multidisciplinary by 14 participants. This study was conducted on benign and malignant patients with chronic liver disease. For this reason, It would be more scientifically and ethically appropriate to have a gastroenterologist or hepatologist in this study.
3) The abstract section of the study is appropriate.
4) Keywords are too many and it is recommended to reduce them. Do not use abbreviations in keywords.
5) The Introduction of the study: The introductory section of the study is long and contains too much theoretical information. It would be more appropriate to remove the following section. For example:
More recently, diagnostic algorithms such as the GALAD score (gender × age × log AFP × PIVKA-II), the ASAP score (age, sex, AFP, PIVKA-II) [27], the GAAD algorithm (a logistic regression model that includes PIVKA-II, AFP, age, and biological sex) [28], and the hepatocellular carcinoma early detection screening (HES) V2.0 (age, AFP, alanine aminotransferase, platelets, AFP-L3, and PIVKA-II) [29] have been proposed. These algorithms are capable of combining various variables, including PIVKA-II and AFP, and demonstrate improved performance in identifying HCC, even in early forms, compared to single biomarkers.
It is sufficient to remove this paragraph.
Materials and Methods:
1) Study Design: This part of the study needs to be written more clearly. The study group includes both primary and recurrent HHC patients. You need to clearly state how you achieved homogeneity here.
2) The time period in which the cross-sectional study begins and ends should be clearly stated.
3) How many times were blood samples taken from the patients? Please indicate this in the study. Which parameters were examined in the blood taken should be clearly stated in the study.
4) EASL guidelines 6, Add a literature number end of the sentence.
​Results
1) There are many etiological factors related to chronic liver disease in the groups. Is there any difference in PIVKA-2 and AFP levels in the control groups according to the etiology of chronic liver disease? Including this topic in your study will add different value.
2) Too many statistical figures are given in the study. This situation makes it difficult for the reader to understand. My opinion is to combine some of them and give them in the form of a table. If this cannot be done, it is appropriate to write the explanations under the table more clearly.​
Discussion
1The discussion and conclusion sections of the study are written clearly and understandably and there is no problem.
Yours sincerely
Prof. Dr. Gokhan Adas
Author Response
Reviewer 2
Dear Authors
The clinical study titled ‘Diagnostic Performance of PIVKA-II in Italian Patients with Hepatocellular Carcinoma’ was evaluated. The study was carried out with scientifically and multidisciplinary. I congratulate all the people who participated in the study. ​My opinions and suggestions regarding the study are given below. The study comply with scientific criteria and is suitable for publication.
Dear Reviewer, thank you for your work, your appreciation, and for providing comments that have helped improve our manuscript.
1) The title of the study is appropriate
Thanks for your comment.
2) The people participating in the study are suitable for team ethics. The study was conducted multidisciplinary by 14 participants. This study was conducted on benign and malignant patients with chronic liver disease. For this reason, It would be more scientifically and ethically appropriate to have a gastroenterologist or hepatologist in this study.
Thank you for your appreciation. The group includes several members with diverse experience, and the hepatologists involved are Prof. Pietro Andreone, international expert with 40 years of experience in the treatment of liver diseases, Giovanni Vitale, Ranka Vukotic, Alessandra Scuteri, Carmela Cursaro, and Filippo Gabrielli.
3) The abstract section of the study is appropriate.
Thanks for your comment.
4) Keywords are too many and it is recommended to reduce them. Do not use abbreviations in keywords.
Thanks for your comment, the change was made as requested.
5) The Introduction of the study: The introductory section of the study is long and contains too much theoretical information. It would be more appropriate to remove the following section. For example:
More recently, diagnostic algorithms such as the GALAD score (gender × age × log AFP × PIVKA-II), the ASAP score (age, sex, AFP, PIVKA-II) [27], the GAAD algorithm (a logistic regression model that includes PIVKA-II, AFP, age, and biological sex) [28], and the hepatocellular carcinoma early detection screening (HES) V2.0 (age, AFP, alanine aminotransferase, platelets, AFP-L3, and PIVKA-II) [29] have been proposed. These algorithms are capable of combining various variables, including PIVKA-II and AFP, and demonstrate improved performance in identifying HCC, even in early forms, compared to single biomarkers.
It is sufficient to remove this paragraph.
We thank you for the suggestion. We removed the paragraph.
Materials and Methods:
1) Study Design: This part of the study needs to be written more clearly. The study group includes both primary and recurrent HHC patients. You need to clearly state how you achieved homogeneity here.
Thank you for the suggestion. Patients were enrolled consecutively. This information has been included in the methods, results and acknowledged as a limitation in the discussion section. As our clinical center is highly specialized in the treatment of HCC, the case series included a relatively high number of patients with HCC compared to its prevalence in the general population or among cirrhotic patients. Homogeneity was ensured by strictly adhering to the study's inclusion and exclusion criteria. We also add the data in the result section: “Considering the 80 patients with HCC, 29 cases were newly diagnosed (de novo), while 51 were recurrent cases.”
2) The time period in which the cross-sectional study begins and ends should be clearly stated.
Thank you. We add the sentence “The enrollment period began on July 1, 2017, with the inclusion of the first patient, and concluded on July 16, 2019, with the enrollment of the final patient.”
3) How many times were blood samples taken from the patients? Please indicate this in the study. Which parameters were examined in the blood taken should be clearly stated in the study.
We thank the reviewer for pointing out that our description of the methods may have caused confusion. As this is a cross-sectional study, the blood samples were collected at a single time point. We have made the necessary clarification in the "Samples Collection and Markers Quantification" section of the Materials and Methods
4) EASL guidelines 6, Add a literature number end of the sentence.
Thank you for bringing this to our attention; we have made the necessary correction.
​Results
1) There are many etiological factors related to chronic liver disease in the groups. Is there any difference in PIVKA-2 and AFP levels in the control groups according to the etiology of chronic liver disease? Including this topic in your study will add different value.
Thank you for raising this point. We found it useful to add this information to Table 1; however, we decided not to perform this analysis because the number of patients for each specific etiology would not be statistically significant for comparison. Moreover, the difference between HCC and controls is so substantial that we believe the current analysis is sufficient.
2) Too many statistical figures are given in the study. This situation makes it difficult for the reader to understand. My opinion is to combine some of them and give them in the form of a table. If this cannot be done, it is appropriate to write the explanations under the table more clearly.​
Thank you for your comment. To answer this comment and another reviewer's comment we created a new table indicated as Table3.
Discussion
1The discussion and conclusion sections of the study are written clearly and understandably and there is no problem.
Thanks for your comment.
Yours sincerely
Prof. Dr. Gokhan Adas
Reviewer 3 Report
Comments and Suggestions for Authors
This is a study of PIVKA-II as a diagnostic in hepatocellular carcinoma. Overall, the study results may be of interest to readers. Some comments are given below.
Major comments
Some motivation for the study design is needed, specifically, for (1) why the liver cirrhosis and chronic hepatitis C were chosen as the two controls, and (2) also why the quantities 80/111/111were chosen. Are these quantities reflective of prevalence, or of what was available?
Table 1: There are misalignment problems with the rightmost column. Some rows there do not line up with the row in the other columns.
Figure 1: Please verify that the p-values in this figure are rank-based, and not t-test based. The data are highly skewed, so rank-based nonparametric methods should be used.
Figure 2: Please verify that the p-values in this figure are rank-based, and not t-test based. The data are highly skewed, so rank-based nonparametric methods should be used.
Figure 3: Please verify that the p-values in this figure are rank-based, and not t-test based. The data are highly skewed, so rank-based nonparametric methods should be used.
Figure 5: For these data, an F-test may be sensitive to the outliers. Suggest using a nonparmetric Kruskal-Wallis test instead.
Page 9: “PPV=54.70%” and “NPV=91.35%”. The PPV and NPV will be functions of the prevalence of the disease, which in this case was manipulated by the investigators since they shoes the sample sizes 80/111/111 in each group. Therefore, please either justify these proportions, or calculate PPV and NPV using proportions observed in clinical settings.
Figure 7 and Figure 9: If the optimal cutpoint was chosen using these data and then the same dataset used to calculate PPV and NPV, then the data will be overfitted. Ideally, cross-validation should be used here. But the bias here is probably negligible in this setting.
Page 11: “Significantly enhances the diagnostic capability” seemed strong to this reviewer.
Page 13: “The cut-off identified by our study as the optimal…” The cutoff identified as optimal by the Youden index has some issues such as that it will not be optimal for all applications (depends on cost of different type of clinical errors), and it may not be optimal for future studies or patient populations, and there are other indexes that are used for optimization. Suggest making the language more speculative here.
Minor comments
Page 12: “east” should be “easy”
Page 12: “AFT he detection” should be “AFP the detection”
Page 2: “contribuiting” should be “contributing”
Page 5: “F=0.662, p=0.421” When an F-statistic is given, the degrees of freedom should also be given. There should be a numerator degrees of freedom and a denominator degrees of freedom.
Page 6: There is a repeat phrase “with single nodule compared to those with single nodule compared to those with…”
Page 6: “D=369.53” Please state whether this is Cohen’s D, or something else.
Page 7: “R=0.356” Please state whether this is Spearman’s or Pearson’s correlation, or something else.
Page 8: “F=0.023” Please provide the numerator and denominator degrees of freedom for the F-test.
Author Response
Reviewer 3
Comments and Suggestions for Authors
This is a study of PIVKA-II as a diagnostic in hepatocellular carcinoma. Overall, the study results may be of interest to readers. Some comments are given below.
Thank you for your appreciation.
Major comments
1) Some motivation for the study design is needed, specifically, for (1) why the liver cirrhosis and chronic hepatitis C were chosen as the two controls, and (2) also why the quantities 80/111/111were chosen. Are these quantities reflective of prevalence, or of what was available?
Thank you for raising this point. We did not calculate the sample size based on statistical power but rather on an estimate of the patient cases referred to our operational unit, as is typical in an exploratory study. This is because, at the time the study was designed in 2017, there were not many studies available on PIVKA-II. As our clinical center is highly specialized in the treatment of HCC, the case series included a relatively high number of patients with HCC compared to its prevalence in the general population or among cirrhotic patients.
Table 1: There are misalignment problems with the rightmost column. Some rows there do not line up with the row in the other columns.
Thank you for your comment. We fixed the table.
Figure 1: Please verify that the p-values in this figure are rank-based, and not t-test based. The data are highly skewed, so rank-based nonparametric methods should be used.
Thank you for your comment. To answer this comment and another reviewer's comment we create a new table, indicated as Table 3.
Figure 2: Please verify that the p-values in this figure are rank-based, and not t-test based. The data are highly skewed, so rank-based nonparametric methods should be used.
Thank you for your comment. To answer this comment and another reviewer's comment we create a new table, indicated as Table 3.
Figure 3: Please verify that the p-values in this figure are rank-based, and not t-test based. The data are highly skewed, so rank-based nonparametric methods should be used.
Thank you for your comment. To answer this comment and another reviewer's comment we create a new table, indicated as Table 3.
Figure 5: For these data, an F-test may be sensitive to the outliers. Suggest using a nonparmetric Kruskal-Wallis test instead.
Thank you for your comment, we performed Kruskal-Wallis test, H score and p value are included in the text.
Page 9: “PPV=54.70%” and “NPV=91.35%”. The PPV and NPV will be functions of the prevalence of the disease, which in this case was manipulated by the investigators since they show the sample sizes 80/111/111 in each group. Therefore, please either justify these proportions, or calculate PPV and NPV using proportions observed in clinical settings.
Thank you for your comment. Given that our clinical center is a national reference point in Italy for the treatment of liver cirrhosis and HCC, we conducted consecutive enrollment of patients admitted to the hospital. This approach resulted, by chance, in a population distributed as follows (80/111/111), a distribution that reflects the prevalence observed in our clinical practice rather than in the general population. We have clarified this point in the Methods section, and we hope it is now clearer to the reader.
Figure 7 and Figure 9: If the optimal cutpoint was chosen using these data and then the same dataset used to calculate PPV and NPV, then the data will be overfitted. Ideally, cross-validation should be used here. But the bias here is probably negligible in this setting.
We thank the reviewer for the comment. We know that this type of analysis being taken from the same sample can lead to overfitting. We have created these two figures to allow the reader a more immediate and easier reading of the data. We remain available to remove the other figures as well, should they prove to be confusing. Additionally, following the suggestion of Reviewer 1, we performed a further analysis using an AFP cut-off of 20 ng/ml, as recommended in the literature, but this resulted in worse performance. Since we had to include this new analysis, we believe that Figures 7 and 9 may be useful.
Page 11: “Significantly enhances the diagnostic capability” seemed strong to this reviewer.
Thank you, we have removed the adverb "significantly."
Page 13: “The cut-off identified by our study as the optimal…” The cutoff identified as optimal by the Youden index has some issues such as that it will not be optimal for all applications (depends on cost of different type of clinical errors), and it may not be optimal for future studies or patient populations, and there are other indexes that are used for optimization. Suggest making the language more speculative here.
Thank you, we have revised the sentence according to your comment: “The cut-off identified in our study as the optimal threshold for detecting HCC, as determined by the Youden index, is slightly higher than that proposed by Asian guidelines. However, it is important to note that the Youden index, while widely used, has its limitations. It may not provide an optimal threshold for all clinical applications, as the choice of cut-off can depend on the relative costs of different types of clinical errors. Additionally, the identified threshold may not necessarily generalize to future studies or different patient populations. Other optimization methods or indexes could yield alternative thresholds that may be better suited for specific contexts.”
Minor comments
Page 12: “east” should be “easy” Thank you.
Page 12: “AFT he detection” should be “AFP the detection” Thank you.
Page 2: “contribuiting” should be “contributing” Thank you.
Page 5: “F=0.662, p=0.421” When an F-statistic is given, the degrees of freedom should also be given. There should be a numerator degrees of freedom and a denominator degrees of freedom.
Thank you for your comment, based on the comments received from other reviewers we performed new analysis and used a new test.
Page 6: There is a repeat phrase “with single nodule compared to those with single nodule compared to those with…” Thank you.
Page 6: “D=369.53” Please state whether this is Cohen’s D, or something else.
Thank you for your comment, based on the comments received from other reviewers we performed new analysis and used a new test.
Page 7: “R=0.356” Please state whether this is Spearman’s or Pearson’s correlation, or something else.
We specified the type of the variable
Page 8: “F=0.023” Please provide the numerator and denominator degrees of freedom for the F-test.
Thank you for your comment, based on the comments received from the other reviewers we performed new analysis and used a new test. We will keep your input in mind for future work.
Round 2
Reviewer 3 Report
Comments and Suggestions for Authors
The authors have addressed my concerns.